# The Transcriptomic Landscape of Pediatric Astrocytoma

**DOI:** 10.3390/ijms232012696

**Published:** 2022-10-21

**Authors:** Abrahan Hernández-Hernández, Tayde López-Santaella, Aranxa Torres-Caballero, Amarantha Serrato, Ulises Torres-Flores, Diego Montesinos-Valencia, Fernando Chico-Ponce de León, Vicente González-Carranza, Samuel Torres-García, Rosa Rebollar-Vega, Inti Alberto De la Rosa-Velázquez, Rosario Ortiz, Monserrat Pérez-Ramírez, Normand García-Hernández, Antonio García-Méndez, Francisco Arenas-Huertero

**Affiliations:** 1Biología de Células Individuales (BIOCELIN), Hospital Infantil de México Federico Gómez, Ciudad de México 06720, Mexico; 2Laboratorio de Investigación en Patología Experimental, Hospital Infantil de México Federico Gómez, Ciudad de México 06720, Mexico; 3Departamento de Neurología, Hospital Infantil de México Federico Gómez, Ciudad de México 06720, Mexico; 4Laboratorio de Genómica, Red de Apoyo a la Investigación, Universidad Nacional Autónoma de México, Instituto Nacional de Ciencias Médicas y Nutrición Salvador Zubirán, México City 04510, Mexico; 5Laboratorio de Microscopía Electrónica, Facultad de Ciencias, Universidad Nacional Autónoma de México, Ciudad de México 04510, Mexico; 6Unidad de Investigación Médica en Genética Humana, Centro Médico Nacional Siglo XXI, IMSS, Hospital de Pediatría, Ciudad de México 06720, Mexico; 7Centro Médico Nacional, La Raza, IMSS, Hospital General Gaudencio González Garza, Ciudad de México 04510, Mexico

**Keywords:** pediatric astrocytoma, transcriptome, RNA-seq

## Abstract

Central nervous system tumors are the most common solid neoplasia during childhood and represent one of the leading causes of cancer-related mortality. Tumors arising from astrocytic cells (astrocytomas) are the most frequently diagnosed, and according to their histological and pathological characteristics, they are classified into four categories. However, an additional layer of molecular classification considering the DNA sequence of the tumorigenesis-associated genes *IDH1/2* and *H3F3A* has recently been incorporated into the classification guidelines. Although mutations in *H3F3A* are found exclusively in a subtype of grade IV pediatric astrocytoma, mutations in *IDH1/2* genes are very rare in children under 14 years of age. The transcriptomic profiles of astrocytoma in adults and children have been extensively studied. However, there is scarce information on these profiles in pediatric populations considering the status of tumorigenesis-associated genes. Therefore, here we report the transcriptomic landscape of the four grades of pediatric astrocytoma by RNA sequencing. We found several well-documented biological functions associated with the misregulated genes in the four grades of astrocytoma, as well as additional biological pathways. Among the four grades of astrocytoma, we found shared misregulated genes that could have implications in tumorigenesis. Finally, we identified a transcriptional signature for almost all grades of astrocytoma that could be used as a transcription-based identification method.

## 1. Introduction

Central nervous system (CNS) tumors constitute approximately 25% of childhood neoplastic diseases and represent one of the leading causes of cancer-related deaths in children and young adults [1,2,3]. Gliomas arise from glial precursor cells that are present in the brain and spinal cord. These gliomas are named according to their assumed clinicopathological and histological subtype. Astrocytoma is the most common type of glioma diagnosed in children and can occur anywhere in the CNS. According to the World Health Organization in 2007, astrocytomas are classified as low-grade (grades I and II) or high-grade (grades III and IV) astrocytoma (LGA and HGA, respectively) [4,5,6]. Pediatric low-grade astrocytoma (pLGA) is a highly heterogeneous collection of entities representing 25% to 30% of all CNS tumors. The most common subtypes are pilocytic astrocytoma (grade I) followed by diffuse and pilomyxoid astrocytoma (grade II) [6]. Pediatric high-grade astrocytoma (pHGA) includes anaplastic astrocytoma (grade III) and glioblastoma (grade IV), both malignant, diffuse, infiltrating astrocytic tumors, and with a poor prognosis [3,4,7].

On the basis of deep molecular characterization of these tumors, an additional layer of classification has been suggested. In 2016, the WHO classification criteria maintained the grading of astrocytomas and incorporated genetic factors associated with tumorigenesis. Consequently, a reorganization of the previous classification has been developed. For example, diffuse astrocytoma (grade II), anaplastic astrocytoma (grade III), and glioblastoma (grade IV) are now clustered in the diffuse astrocytoma category with the addition of the genetic status (mutant or wild-type) of genomic regions believed to be key for the tumorigenesis (within the *IDH1/IDH2* and *H3F3A* genes). The historically ambiguously classified diffuse intrinsic pontine glioma (DIPG), an infiltrative midline pediatric high-grade glioma (pHGG) with predominantly astrocytic differentiation, has now been included in the diffuse astrocytoma category and named diffuse midline glioma, H3K27M mutant. Meanwhile, pilocytic and pilomyxoid astrocytoma are now part of the other astrocytoma category [4]. 

Although pediatric and adult HGA are indistinguishable under a microscope, they are different molecular entities. In pHGA, classic driver mutations in *IDH1/2* are rare, affecting mainly adolescents [8], indicating that other genetic and epigenetic events are involved in pediatric tumorigenesis. Localization, age of occurrence, presence/absence of typical driver mutations, and neuroanatomical specificity of tumor subtypes suggest that unique biological factors, such as developmental constraints, cell origin, and tumor microenvironment, may contribute to specific forms of tumorigenesis in pediatric tumors [8]. Therefore, to study the biology, clinical features, and other aspects of tumorigenesis, pediatric astrocytoma should be considered as different from its counterpart in adults. Furthermore, even though classical driver mutations are rare in pediatric astrocytoma, these should be screened and considered when studying these tumors. In this study, we analyzed the transcriptome of pLGA and pHGA. We found several well-documented and additional biological functions associated with the misregulated genes in the four grades of astrocytoma. In addition, we found shared misregulated genes that could have implications in the tumorigenesis. Finally, we identified a transcriptional signature of unique markers for nearly all grades of astrocytoma that could be used as a transcription-based identification method for the most common pediatric astrocytoma.

## 2. Results

### 2.1. Clustering of Astrocytoma Transcriptomic Profiles according to Their Grade of Malignancy

To determine the transcriptomic profiles of pediatric astrocytomas, we sequenced total RNA (RNA-seq) from 15 biopsies of pediatric astrocytomas from the cerebrum and cerebellum, and 5 healthy matching tissues. We followed standard bioinformatic tools for the analysis of the RNA-seq data [9] as described in Figure 1. We performed quality control of the reads observing acceptable total read counts and quality scores for all the sequences (Appendix A). We aligned and mapped the reads to the human reference genome, obtaining between 82–90% of uniquely mapped reads, 1–7% of reads mapped to ribosomal RNA, and 0.5–2% of reads mapped to nonchromosomal RNA (e.g., mitochondrial, sequence contigs not yet mapped on chromosomes) (Appendix A). The obtained percentages are expected for libraries of good quality [10]. Finally, to eliminate composition biases, we performed filtering of genes with low expression and library normalization (Appendix A).

To evaluate how astrocytomas clustered based on their transcriptional profiles, we performed an unsupervised principal component analysis of the RNA-seq data (mapped reads). We observed that astrocytomas clustered according to their low or high grade of malignancy (WHO classification 2007) (Figure 2a) rather than according to their classification based on the absence of causal mutations (i.e., diffuse and other astrocytic tumors, WHO classification 2016) (Figure 2b). It is relevant to mention that all diffuse astrocytic tumors analyzed in this study did not harbor any of the reported mutations in the *IDH1/2* and *H3F3A* genes (Appendix A). Among pLGA, only grade I astrocytoma (pilocytic astrocytoma or PA or A.I) from the cerebellum clustered, while a clear separation between PA in the cerebrum and grade II astrocytoma (diffuse and pilomyxoid astrocytoma, A.II) was not observed (Figure 2a). Within the pHGA, we noticed a separation between grade III astrocytoma (anaplastic astrocytoma, A.III) and grade IV (glioblastoma or GBM, A.IV) (Figure 2a). Finally, we performed unsupervised hierarchical clustering based on the 500 most variable genes among samples and observed again that astrocytomas clustered as low- and high-grade with a clear separation of PA in the cerebellum, anaplastic astrocytoma, and glioblastoma (Figure 2c). Therefore, the transcriptomic profiles of pediatric astrocytoma are related to their grade of malignancy.

### 2.2. Transcriptomic Profile and Altered Biological Functions in Pilocytic Astrocytoma

We then evaluated differentially expressed genes (DEGs) between each grade of astrocytoma and matched healthy tissues (Appendix A). We did not observe DEGs in PA from the cerebrum even after the fold change threshold was removed. However, when we compared PA in the cerebellum versus healthy cerebellum (FC > 1.5, FDR < 0.05), we found 763 and 788 genes up- and downregulated, respectively (Figure 3a and Appendix A). To identify differences in PA from the cerebrum and cerebellum, we compared their transcriptomes and found no DEGs between these two regions of the brain. However, using our RNA-seq data to deduce gene fusions, we found that only PAs from the cerebellum express the most typical *BRAF-KIAA1549* gene fusion [11] (Figure 3b).

To gain a better understanding of the heterogeneity of misregulated genes, we analyzed their nature. We found that most of the up- and downregulated genes (84 and 71%, respectively) correspond to protein-coding genes. The rest represented ncRNAs (noncoding RNAs), pseudogenes, snoRNAs (small nucleolar RNAs), snRNAs (small nuclear RNAs), and others (i.e., novel genes/transcripts, putative genes, intronic sense/antisense transcripts, antisense transcripts, etc.) (Figure 3a and Appendix A). Throughout functional annotation clustering using DAVID analysis (Database for Annotation, Visualization, and Integrated Discovery) [12], we searched biological functions of the protein-coding misregulated genes and selected the top 10 annotation clusters (Appendix A). Among the misregulated protein-coding genes, we found several common biological pathways that have previously been reported to be over- and underexpressed in these tumors (asterisks in Appendix A), including positive regulation of MAP kinase activity (Appendix A), the most frequent dysregulated pathway in pediatric cerebellar PA [3,13,14,15]. Finally, when we compared the up- and downregulated genes in our experimental setup with misregulated genes reported by microarray profiling of PA (from both the pediatric and adult population) [16,17], we found a wide overlap between the datasets (Appendix A). Collectively, the similarities of altered biological pathways and genes between this and other studies support our data and suggest that the cerebellar pediatric PA transcriptome is quite particular. 

### 2.3. Grade II Astrocytoma 

Although rare, grade II astrocytomas are present throughout major brain and central nervous system structures with a higher frequency in the cerebrum (frontal and temporal lobes) [5,18]. In adults, diffuse astrocytomas are nearly all characterized by mutations in *IDH* genes. However, these mutations are not present in the pediatric population [7,19]. Accordingly, we did not detect the typical mutations in *IDH1/2* and *H3F3A* genes in grade II astrocytoma [20]. We then evaluated the transcriptional changes in grade II astrocytoma with various locations compared to matching tissues and locations (Appendix A) and found 84 and 6 up-and downregulated genes, respectively (Figure 4a and Appendix A). Among the biological pathways associated with the upregulated genes, we found very similar annotations to those found for PA (Appendix A). For the downregulated genes, we did not find biological functions clustered by their annotations. However, analysis of the annotations of the individual genes revealed that their roles are very similar to those found in the altered biological pathways in PA (Appendix A). In line with these similitudes, we observed that most of the upregulated genes in grade II astrocytoma (81 out of 84) were also upregulated in PA (Figure 4b). Furthermore, all six downregulated genes in grade II astrocytoma are also downregulated in PA (Figure 4b). These data suggest that A.II and PA show similar altered biological pathways. We then looked at the annotations for the three genes that are upregulated only for A.II and found that the p53 signaling pathway (for *TP53I3*) and intracellular trafficking (for *SNX22*) are specific for this type of astrocytoma (Figure 4b).

### 2.4. Pediatric High Grade Astrocytoma—Anaplastic Astrocytoma

The pHGA group includes anaplastic astrocytoma (grade III) and glioblastoma (grade IV), both malignant, diffuse, and infiltrating astrocytic tumors [4]. These two subtypes are grouped into diffuse astrocytomas based on their driver mutations. However, as expected in our experimental setup, we have pHGA of the cerebellum and cerebrum without mutations in *IDH1/2* and *H3F3A*. We noticed that their transcriptomes are more similar according to their grade of malignancy (i.e., grade III and grade IV) (Figure 2a). We therefore analyzed their transcriptomes separately. Although anaplastic astrocytoma is more commonly found in cerebrum, in our experimental setup we had two samples of these tumor located in cerebellum, so, we analyzed their transcriptome differences versus healthy cerebellum (Appendix A). We found 21 and 32 up- and downregulated genes, respectively (Figure 4c and Appendix A). Since we found few misregulated genes, instead of searching for annotation clusters, we rather searched for individual annotations (in the KEGG_PATHWAY annotations from DAVID). For the upregulated genes, we found biological functions such as the AMPK/MAPK signaling pathway and endocytosis/cell adhesion molecules, whereas for the downregulated genes, we found the MAPK signaling pathway, microRNAs in cancer, and the Notch signaling pathway, among others (Appendix A).

### 2.5. Pediatric High Grade Astrocytoma—Glioblastoma 

We then compared the glioblastoma transcriptome in the cerebrum versus healthy cerebrum (Appendix A) and found that 540 genes were upregulated and 949 genes were downregulated (Figure 5a and Appendix A). We next decided to investigate the altered biological pathways in these types of tumors (Appendix A). Consistent with other reports describing increased mitotic activity in pediatric glioblastoma [4,5,7,21], we found that the most significant annotation group has terms such as sister chromatid cohesion, centromere, and kinetochore. Furthermore, subclustering of this top cluster shows annotations relevant for the biology of pediatric glioblastoma such as mitotic nuclear division, spindle microtubule and protein phosphorylation, mitotic spindle assembly checkpoint, regulation of chromosome segregation, and cell cycle (Appendix A). To support our RNA-seq and DEG data, we performed RT-qPCR of two randomly downregulated genes from our analysis (Figure 5b). We evaluated their expression using five paraffin-embedded pediatric glioblastomas (with WT genotype for *IDH1/2* and *H3F3A* genes) and two paraffin-embedded healthy cerebrums. Our results indicated that the average relative expression of these two genes in the glioblastoma biopsies was negatively regulated, although one of the genes did not pass the significance threshold (Figure 5c). Nevertheless, this overall downregulation and RNA-seq data suggest data correlation.

Although histologically different, pediatric astrocytic tumor grades III and IV are considered a single category for therapeutic purposes [7,21]. To correlate these histological differences with their transcriptional profiles, we compared misregulated genes shared between grade III and IV astrocytoma (Figure 5d). We found that shared upregulated genes have annotations associated with mitotic chromosome condensation and cell division, among others (Appendix A), while shared downregulated biological functions have annotations related to synapse/cell junction and cytoplasmic vesicles, among others (Appendix A). Regarding the unique genes, we observed that genes upregulated in anaplastic astrocytoma are associated with the MAPK/ AMPK signaling pathways, among others (Appendix A). Meanwhile, downregulated genes unique for anaplastic astrocytoma have annotations such as cell division, MAPK signaling pathway, cellular response to DNA damage stimulus, among others (Appendix A). These data strongly correlate with the histological similarities/differences observed between these two subtypes of high-grade pediatric astrocytoma and could be helpful in understanding their biology.

### 2.6. Similarities among All Grades of Astrocytoma Reveal a Putative General Tumorigenesis Factor 

We then looked for genes that were upregulated and shared in all grades of astrocytoma and found two. One of them is the annotated and curated coding protein gene with functional annotations related with protein glycosylation on the Golgi membrane (Figure 6a). For downregulated genes, only one gene was found to be shared among astrocytomas. This feature is the long noncoding gene *linc-OIP5* (as known as Cyrano) (Figure 6b). To corroborate that Cyrano expression was decreased in our samples, we analyzed the log2 and normalized counts from the RNA-seq data and observed lower read counts in all astrocytoma samples compared to healthy cerebrum and cerebellum (Figure 6c). Therefore, our analysis of RNA-seq may have underscored genes with potential implications in the tumorigenesis of pediatric astrocytoma.

### 2.7. Unique Upregulated Genes in each Grade of Astrocytoma May Function as a Transcriptional Signature to Classify These Tumors 

The search for gene expression signatures (uniquely expressed genes) is a useful clinical and laboratory tool used to produce customized panels for the identification, stratification and/or classification of specific subtypes of cancer tumors [22,23,24,25,26,27,28,29,30,31,32]. Thus, we searched for the top genes that were significantly upregulated in every astrocytoma grade. We found 20, 26, and 24 uniquely expressed genes in cerebellar pilocytic, cerebellar anaplastic and glioblastoma astrocytoma, respectively (Figure 7a–c, genes not shown). For diffuse astrocytoma, we did not find significant genes that were uniquely expressed. Furthermore, to distinguish between healthy and cancerous tissues, we searched for uniquely expressed markers in the cerebrum and cerebellum, finding the expression of 22 genes (Figure 7d).

We then made a list of only protein-coding genes from these uniquely expressed genes and produced a signature panel with 68 genes (14, 17, 20, and 17 for cerebellar PA, cerebellar anaplastic and glioblastoma, respectively) and used it to perform unsupervised clustering of all the samples. With this analysis, we observed a clear separation of every grade of pediatric astrocytoma (Figure 8a). We then used this transcriptional signature to produce ssGSEA scores (single-sample gene set enrichment analysis) on our data and projected them onto the publicly available transcriptomic dataset of pediatric astrocytoma. ssGSEA permits to define an enrichment score that represents the degree of absolute enrichment of a gene set in each sample within a given dataset [33,34]. First, we generated a ssGSEA score for our gene signature. Since we selected the unique top-upregulated coding genes for our gene signature, we obtained an absolute enrichment score for our genes on the pediatric astrocytoma transcriptomes, with a clear separation/identification of the four types of tumors and healthy brain. Cerebellar PA, anaplastic astrocytoma, glioblastoma, and healthy brain are clearly clustered according to the transcriptomic profile, while cerebrum PA and grade II astrocytoma are mixed in the same cluster (Appendix A).

Then, we calculated the ssGSEA score of our signature in publicly available transcriptomic profiles of pediatric astrocytoma. We identified a GEO dataset (GSE73066) with 47 transcriptomic profiles of pediatric PAs in different locations of the brain [35]. Since this microarray dataset only has pediatric PA, we expected a high ssGSEA score and a clear identification of cerebellar PA. Furthermore, to refine our cerebellar PA gene signature, we performed a pairwise comparison between cerebellar PA versus each of the other types of astrocytoma and found that only the comparison between cerebellar PA versus glioblastoma revealed 10 differentially expressed genes (FC > 2.0, FDR < 0.05). Then, we added only the coding genes to our PA signature (7 genes). We observed that most of the cerebellar PA with high enrichment scores (of cerebellar PA signature) were clustered together, with few other PA from other locations of the brain (cluster 1 in Figure 8b). We also observed two clusters without enrichment of the cerebellar PA signature. However, these two clusters contain few cerebellar PA and remaining PA from other brain locations (clusters 2–3 in Figure 8b). Although most of the pediatric cerebellar PA are identified with our gene signature, there are some cerebellar PA without the signature; however, we cannot rule out that these cerebellar PAs are different due to the absence of the BRAF-KIAA1549 fusion, which is present in the cerebellar PA used to obtain our signature panel (Figure 3b). Similarly, although we could not obtain information of their location in the brain, in another microarray dataset of 41 pediatric PAs (GSE5675) [36], we observed the identification of a large cluster of cerebellar PAs with our gene signature (Appendix A). Therefore, these analyses suggest a putative panel of markers (protein coding genes) that could be used as transcriptional signature to classify these types of pediatric astrocytoma.

## 3. Discussion

By analyzing the whole transcriptome of different degrees of pediatric astrocytoma, we found that their transcriptional profiles are related to their classification into low- and high-grade, which is directly associated with their malignancy [3,7]. Within the low-grade glioma (pilocytic and grade II), we observed a clear separation of the pilocytic astrocytoma in the cerebellum only. This suggests that while low-grade glioma in the cerebrum are quite heterogenous, pilocytic astrocytoma in the cerebellum have a very distinctive transcriptional profile even from their counterpart in cerebrum. This observation is not unexpected, since other studies have shown that pilocytic astrocytoma in the cerebrum and cerebellum have differential gene expression and DNA methylation patterns [36,37,38]. Furthermore, only cerebellar PA shows the chimeric transcript BRAF-KIAA1549, suggesting that this is the underlining reason for their unique transcriptional profile despite histological similarity with PA from the cerebrum cortex. The clustering of anaplastic astrocytoma and glioblastoma within high-grade tumors was also evident, suggesting that although they are sometimes considered a single identity, they are clearly different.

### 3.1. Altered Pathways in Pediatric Astrocytoma

To obtain an overview of the altered biological pathways in different grades of pediatric astrocytoma, we analyzed the associations of the annotation of the coding protein genes and found several well-documented altered pathways for every grade. Although we did not further analyze the possible biological pathways associated to the misregulated noncoding RNAs, we reason that this type of analysis could provide a full overview of the alteration/regulation of the noncoding transcriptome in the different grades of astrocytoma.

### 3.2. Low-Grade Astrocytoma

We found several biological pathways to be altered in PA in the cerebellum, which have been previously described. For instance, studies comparing PA from cerebellum versus healthy cerebellum found immune system-related genes to be upregulated [39,40]. We found immunity and innate immune responses as the most significantly upregulated biological pathways in PA of the cerebellum. Other studies have found that genes involved in neurogenesis, synaptic transmission, central nervous system development, and potassium ion transport were significantly negatively regulated in pediatric PA compared to the normal cerebellum [41]. Similarly, within the downregulated genes in our study, we found synapse and cell junction, neurotransmitter-gated ion channel, and neurogenic differentiation factor among the top annotation clusters. Furthermore, the most frequent dysregulated pathway in PA in the cerebellum is the positive regulation of MAP kinase activity [3,13,14,15]. Indeed, we also found six upregulated genes associated with the positive regulation of MAP kinase activity. Additionally, we underscored other genes and biological pathways that could be exclusive of the pediatric population and could potentially help to understand the biology of pediatric PA.

We also noticed that most of the misregulated genes in grade II astrocytoma are the same as in pilocytic astrocytoma, reinforcing the fact that both subtypes are of low-grade malignancy. However, from the few specific upregulated genes in grade II astrocytoma, we identified *TP53I3,* which is involved in the p53 signaling pathway. Although molecularly different, in diffuse astrocytoma from adults, the p53 signaling pathway has also been reported to be important for early events in tumor formation [7,18,19]. Thus, this could explain some of the histological similitudes found between adult and pediatric grade II astrocytoma [18].

### 3.3. High-Grade Astrocytoma

Pediatric and adult high-grade astrocytomas are indistinguishable under a microscope, but they are distinct molecular entities. In pediatric tumors, driver mutations in *IDH1/2* are rare; instead they harbor specific recurrent mutations in the genes encoding H3.3 (*H3F3A*) and H3.1 (*HIST1H3B*, *HIST1H3C*) histone variants, resulting in amino acid substitutions in two key residues in the histone tail: lysine-to-methionine at position 27 (K27M) and glycine-to-arginine or -valine at position 34 (G34R/V) [42,43]. Currently, the genetics and epigenetics of pHGA harboring mutations in *IDH1/2* and *H3F3A* are under intensive study [3,21,42,44]. In our experimental setup, we had pediatric high-grade cerebellum and cerebrum astrocytoma without mutations in *IDH1/2* and *H3F3A*. This was not unexpected, since mutations in H3K27 are mostly found in high-grade astrocytoma in the brainstem and only a small percentage of pHGGs in the cerebrum cortex display mutations in H3G34. Therefore, our transcriptomic profiling of pHGA in cerebellum (grade III, *IDH1/2* and *H3F3A* wild-type) and cerebrum (grade IV, *IDH1/2* and *H3F3A* wild-type) may greatly contribute to the understanding of these tumor subtypes.

### 3.4. Cyrano, a lncRNA That May Be Involved in the Tumorigenesis of Pediatric Astrocytoma

We identified a long noncoding gene known as Cyrano. This gene is downregulated in every pediatric astrocytoma regardless their grade of malignancy. Cyrano has a high expression in the nervous system and is important for controlling neurogenesis during development [45]. The up- and/or downregulated expression of this gene has been broadly reported in several human cancer tumors [8]. Additionally, Cyrano has been reported to be upregulated in glioma tissue and cell lines of adult patients [46]. Considering that astrocytomas in adults and children are molecularly different and that Cyrano displays both oncogenic and tumor-suppressive roles, it is possible that Cyrano might be playing opposite roles in the tumorigenesis of pediatric versus adult astrocytoma.

### 3.5. A Transcriptional Signature to Classify Pediatric Astrocytoma 

Finally, we identified a grade-specific transcriptional signature that may be useful for classifying these subtypes of pediatric tumors. Consistent with this, mRNA expression-based subtyping of breast cancer [30,31,32] has envisioned the creation of the 21-gene OncotypeDx assay (Genome Health Inc, Redwood City, CA, USA). This technique can be used to stratify the risk of early-stage estrogen receptor (ER)-positive breast cancer [28,29]. Furthermore, gene expression of 50 genes (PAM50) [27] is also used as a risk predictor of breast cancer based on intrinsic subtypes [22,23,24,25,26,27]. Many studies have reported transcriptional signatures for pediatric pilocytic astrocytoma related to their location in the brain. However, comparisons have been made directly between PAs at different locations and sometimes mixing pediatric and adult samples [16,35,36,47]. In this study, by including healthy pediatric brain tissues and the four histopathological grades of astrocytoma (with their molecular classification), we have produced a panel of genes that can be used to identify pediatric astrocytoma. However, more robust validations using similar grades of pediatric astrocytoma in larger cohorts may provide a more compelling reliability of this transcriptional signature. Therefore, in addition to considering the unified molecular and histopathological classification criteria (WHO classifications 2006 and 2017), we suggest that the panel of unique coding protein genes reported in this work could be useful for the characterization of classic subsets of pediatric astrocytoma throughout mRNA and/or protein expression.

Although we have strengthened our dataset by replicating the observations about altered biological pathways, overlapping of misregulated genes with other genome-wide genomic studies and rt-qpcr, we acknowledge the caveat of the low sample numbers and recommend to further validate our observations before jumping into more generalized and mechanistic studies.

## 4. Materials and Methods

### 4.1. Human Samples

Patients newly diagnosed with astrocytoma in the cerebrum and cerebellum were recruited from the Hospital Infantil de México Federico Gómez and Hospital de Pediatría, Centro Médico Nacional Siglo XXI. Tumor biopsies were collected according to International and Institutional Guidelines from children and adolescents younger than 18 years of age and diagnosed with astrocytoma before any treatment. Healthy control tissues were collected, either from autopsies of patients deceased from non-astrocytoma causes or from surgical procedures that involved the removal of healthy tissue to reach other brain tumors (not related to astrocytoma), according to International and Institutional Guidelines. A summary of clinicopathological features of patients is depicted in Appendix A.

### 4.2. RNA Extractions and NGS

Tumor biopsies and control tissues were collected and immediately submerged in RNAlater Stabilization Solution (R0901-500ML-PW Sigma or AM7024 Invitrogen). All biopsies were stored at −80 °C until further use. Simultaneous purification of RNA and DNA from tumor and healthy (control) biopsies with the AllPrep DNA/RNA/miRNA Universal kit (QIAGEN) was performed according to the manufacturer instructions. Purified DNA was used to perform Sanger sequencing of driver mutations within *IDH1/2* and *H3F3A* as recommended by the WHO [4]. In a previous study, we reported the genotypic and histopathologic/molecular classification of the samples used in this work [20] (Appendix A). Purified RNA with an acceptable RIN value of 5.5 to 7.6 was used to perform sequencing libraries with the TruSeq Stranded total RNA with Illumina Ribo-Zero Plus rRNA depletion kit (Illumina, Inc. San Diego, CA 92122, USA). Lastly, 2 × 125 paired-end sequencing was performed in a HiSeq2500 (Illumina, Inc. San Diego, CA 92122, USA).

### 4.3. Data Analysis 

The quality control of the raw data files was performed (fastq) with FastQC v0.11.9 (https://www.bioinformatics.babraham.ac.uk/projects/fastqc/, accessed on 16 November 2021) and MultiQC (https://multiqc.info, accessed on 16 November 2021) [48]. The reads were aligned to the human reference genome, ribosomal RNA, and nonchromosomal RNA (e.g., mitochondrial genome, sequence contigs not yet mapped on chromosomes) with STAR 2.7.9a [49]. Sorting of SAM files and BAM files was performed using Samtools 1.13 [50]. These BAM files were used to generate the read summarization and count table, in which the number of reads assigned to each feature in each library was recorded, using featureCounts [51] from the subread 2.0.3 package [52]. The resulting count table was used to perform trimmed mean of M values (TMM) normalization and gene differential expression in the Empirical Analysis of Digital Gene Expression Data in R (EdgeR) from Bioconductor [53,54,55]. Finally, a functional analysis of the mis regulated genes was performed with the Database for Annotation, Visualization and Integrated Discovery (DAVID) v6.8 [56,57]. For the detection of the chimeric transcript BRAF-KIAA1549 from RNA-Seq data, we used the command-line tool Arriba [58]. Finally, to calculate the enrichment scores of the gene set, we used single-sample GSEA (ssGSEA), an extension of Gene Set Enrichment Analysis (GSEA) [59]. Series matrix files for the GSE73066 and GSE5675 datasets (from the Gene Expression Omnibus portal) were downloaded and log-normalized before calculation of the ssGSEA score with the genes from the signature panel. Calculation and visualization of the ssGSEA score were performed using the R-based packages matrixStats [60], circlize [61], Complex heatmaps [62], and data.table (https://github.com/Rdatatable/data.table, accessed on 29 September 2022).

### 4.4. Genotyping

In a previous study, we genotyped the status of *IDH1/2* and *H3FA3* in the samples used for our RNA sequencing assay [20]. For genotyping of paraffin-embedded grade IV astrocytoma, DNA from five different grade IV astrocytoma and two healthy cerebrums was extracted. Paraffin-embedded samples were sectioned and from two to three sections were submerged in 500 μL of xylene and heated at 65 °C in a water bath for 10 min. After incubation, the samples were pelleted by centrifugation at 6780 g for 7 min, and the supernatant was discarded. A second step of centrifugation was repeated. The pelleted tissue was washed with 1 mL of absolute ethanol, followed by another washing step with 1 mL of 70% ethanol. On every wash, the supernatant was eliminated using the previous centrifugation step. The pellet was allowed to air dry for 5 min and then 1 mL of lysis buffer (10 mM Tris HCl pH 8, 0.5% SDS and 5 mM EDTA) with 0.5 mg/mL proteinase K was added. The suspensions were mixed by vortex for 5 sec and incubated at 55 °C overnight in a water bath with continuous shaking. The next day, 500 μL of 5M NaCl were added to the sample, and it was vigorously mixed and pelleted by centrifugation at 6708 g for 15 min. Finally, for DNA precipitation, 900 μL of sample was transferred to a new 1.5 mL tube and 600 μL of pre-chilled isopropanol (at 4 °C) was added, then this was centrifugated at 6708 g for 15 min at 4 °C. After the supernatant was removed, the granulated DNA was washed twice with 70% ethanol. The pellet was allowed to dry with the lid open in an incubator at 50 °C for 15 min, and DNA was resuspended with 30 μL of TE buffer. DNA concentration and purity were read with a spectrophotometer (Genova Nano micro-volume spectrophotometer). DNA from each of the grade IV astrocytoma biopsies and healthy control cerebrum was used for *IDH1*/*IDH2* and *H3K27M* genotyping as previously reported [20].

### 4.5. RT-qPCR

RNA was extracted from the same paraffin-embedded biopsies described above. The paraffin-embedded samples were sectioned, and from two to three sections were submerged in 500 μL of Xylene and heated at 50 °C in a water bath for 10 min. After incubation, the samples were pelleted by centrifugation at 6780 g for 7 min, the supernatant was removed, and a second centrifugation step was performed. The pelleted tissue was washed with 700 μL of absolute ethanol followed by a washing step with 700 μL of 70% ethanol. The pellet was allowed to dry at 37 °C for 5 min in a thermoblock (MS major science) and 1 mL of lysis buffer (10 mM Tris HCl pH 8, 0.5% SDS and 5 mM EDTA) with 0.5 mg/mL proteinase K was added. The suspensions were mixed by vortex for 15 s and incubated at 45 °C overnight in a water bath with continuous shaking. RNA extraction was completed with TRIzol Reagent (Invitrogen, Thermo Fisher Scientific, Waltham, MA 02451, USA) according to the manufacturer’s instructions. Finally, the pelleted RNA was resuspended with 15 μL of DNAse-free water. RNA concentration and purity were measured with a spectrophotometer (Genova Nano micro-volume spectrophotometer). With 200 ng of the purified RNA, the cDNA synthesis was performed using the First Strand cDNA Synthesis Kit (Thermo Scientific). The primers for HIVEP2 and GRIN2B were designed and evaluated by RT-qPCR (Appendix A) and performed qPCR with the Maxima SYBR Green qPCR Master mix kit (Thermo Scientific) according to previous reports [63]. The R-based *pcr* package was used for qPCR data analysis [64].

## Figures and Tables

**Figure 1 ijms-23-12696-f001:**
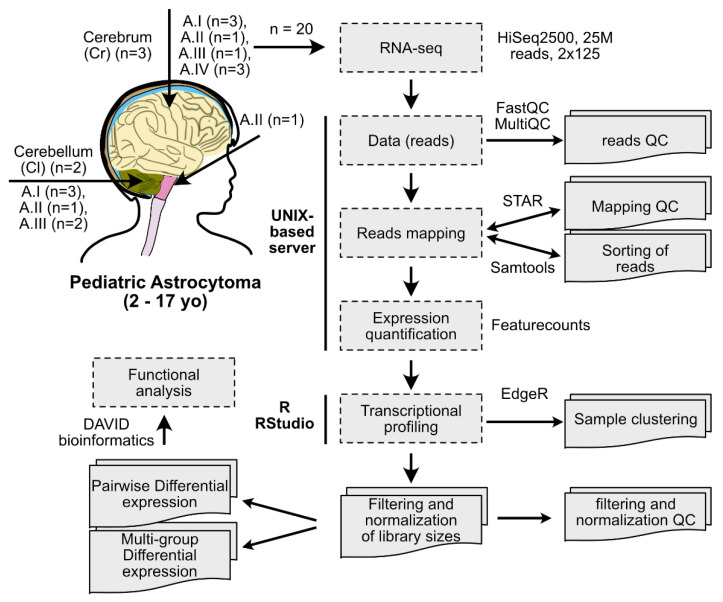
Experimental design of the study. Fifteen pediatric astrocytoma samples and five control tissues were processed for total RNA sequencing (RNA-seq). The bioinformatic pipeline for RNA-seq data analysis is shown.

**Figure 2 ijms-23-12696-f002:**
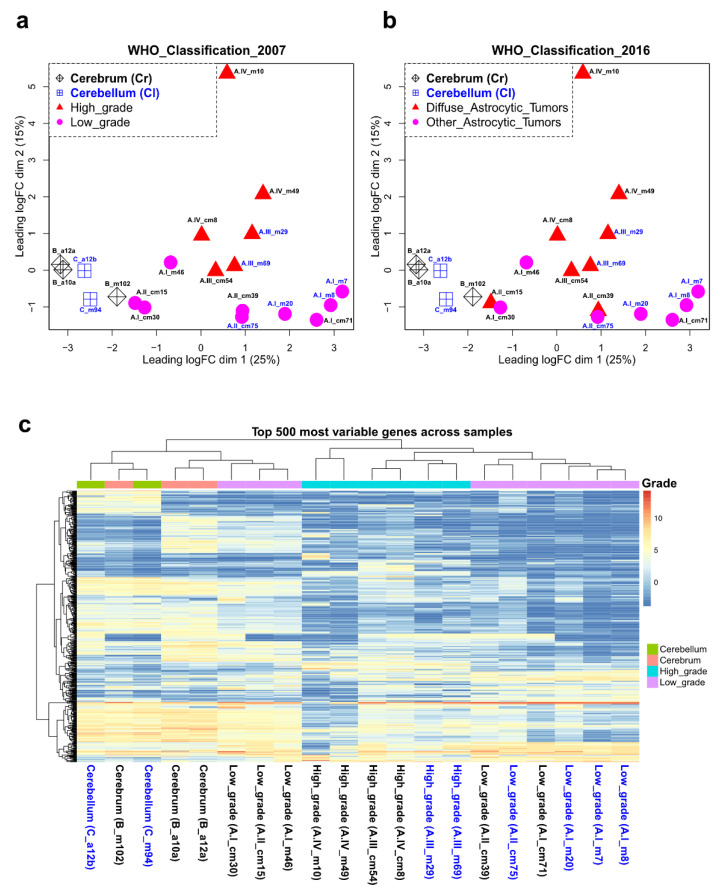
Clustering of astrocytoma transcriptomes. Multidimensional scaling plots displaying the unsupervised principal component analysis of the pediatric astrocytoma RNA-seq data (transcriptomes) classified according to the 2007 WHO criteria (**a**) and 2016 WHO criteria (**b**). (**c**) Heat map displaying the unsupervised hierarchical clustering of the pediatric astrocytoma based on the 500 most variable genes across samples.

**Figure 3 ijms-23-12696-f003:**
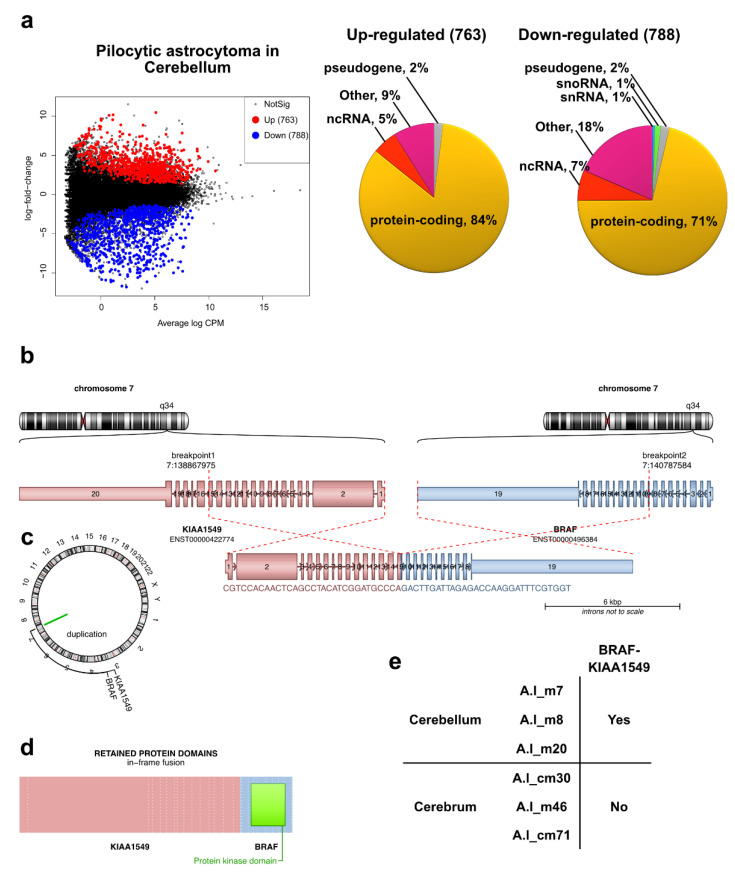
Differential expression of genes in pediatric low-grade astrocytoma. (**a**) Mean difference graphs highlighting the upregulated (red) and downregulated (blue) genes (FC > 1.5, FDR < 0.05) in pilocytic astrocytoma in the cerebellum compared to healthy matching tissue. In the right side of the panels, pie charts displaying the heterogeneity (gene type) of the up- and downregulated genes. (**b**) Structure of the fusion transcript BRAF-KIAA1549 detected in cerebellar PA, derived from the tandem duplication at location 7q34 (**c**). (**d**) Protein domains retained in the fusion protein BRAF-KIAA1549. (**e**) Prescence (Yes) or absence (No) of BRAF-KIAA1549 transcript in PA from different location in the brain.

**Figure 4 ijms-23-12696-f004:**
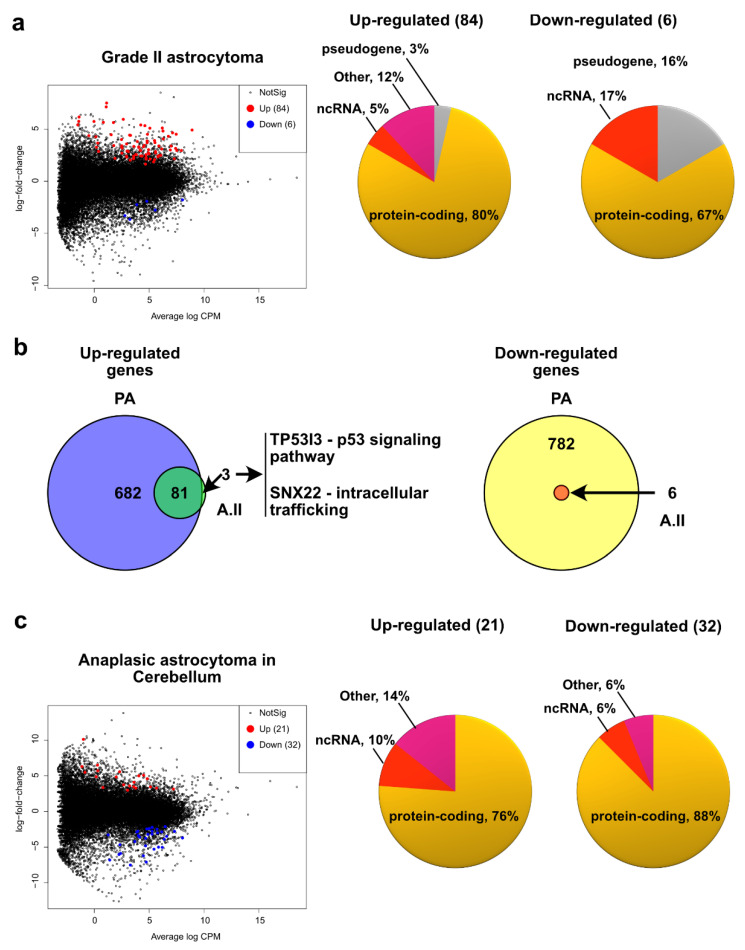
Differential expression of genes in pediatric anaplastic astrocytoma. (**a**) Mean-difference plot highlighting the upregulated (red) and downregulated (blue) genes (FC > 1.5, FDR < 0.05) in grade II astrocytoma compared to healthy matching tissue. In the right side of the panel, pie charts displaying the heterogeneity (gene type) of the up- and downregulated genes. (**b**) Venn diagram showing overlap in the number of upregulated genes (left) and downregulated genes (right) for PA and grade II pediatric astrocytoma. (**c**) Mean-difference plot highlighting the up- and downregulated genes (FC > 1.5, FDR < 0.05) in anaplastic astrocytoma in the cerebellum compared to healthy matching tissue. In the right side of the panel, pie charts displaying the heterogeneity (gene type) of the up- and downregulated genes.

**Figure 5 ijms-23-12696-f005:**
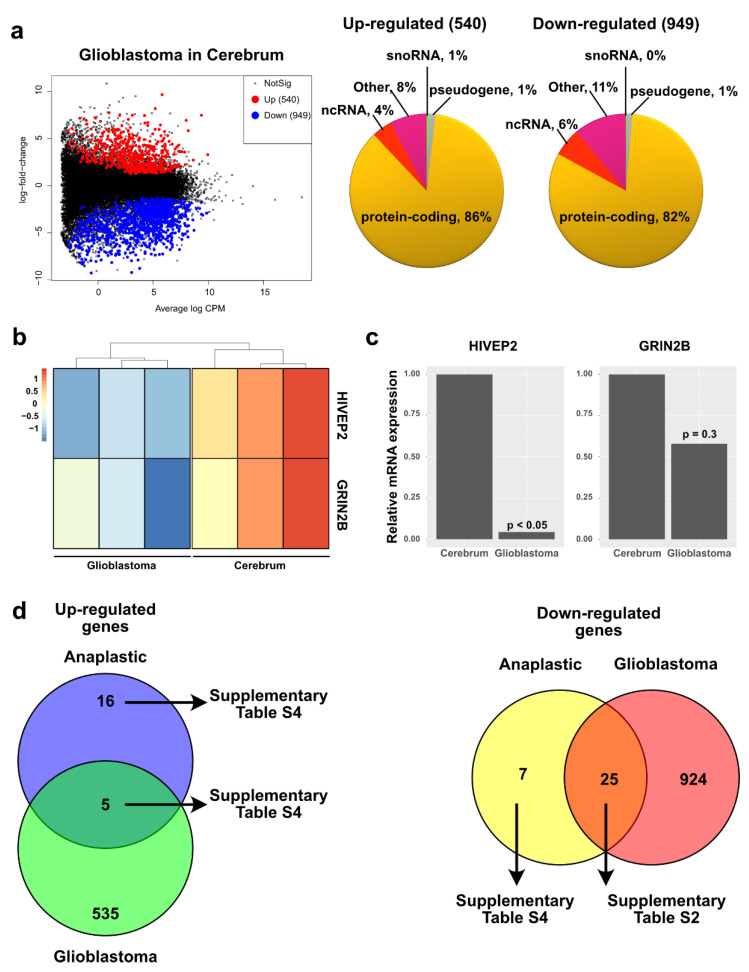
Differential expression of genes in pediatric glioblastoma. (**a**) Mean-difference plot highlighting the upregulated (red) and downregulated (blue) genes (FC > 1.5, FDR < 0.05) in glioblastoma in the cerebrum compared to healthy matching tissue. In the right side, pie charts displaying the heterogeneity (gene type) of the up- and downregulated genes. (**b**) Heat map showing the expression of two randomly picked genes in pediatric glioblastoma and healthy cerebrum from our RNA-seq data. Color bar displays the log count values. (**c**) Relative expression of the genes from panel b in five different samples of pediatric glioblastoma, evaluated by RT-qPCR. (**d**) Venn diagram showing overlap in the number of upregulated genes (left) and downregulated genes (right) for anaplastic astrocytoma and glioblastoma.

**Figure 6 ijms-23-12696-f006:**
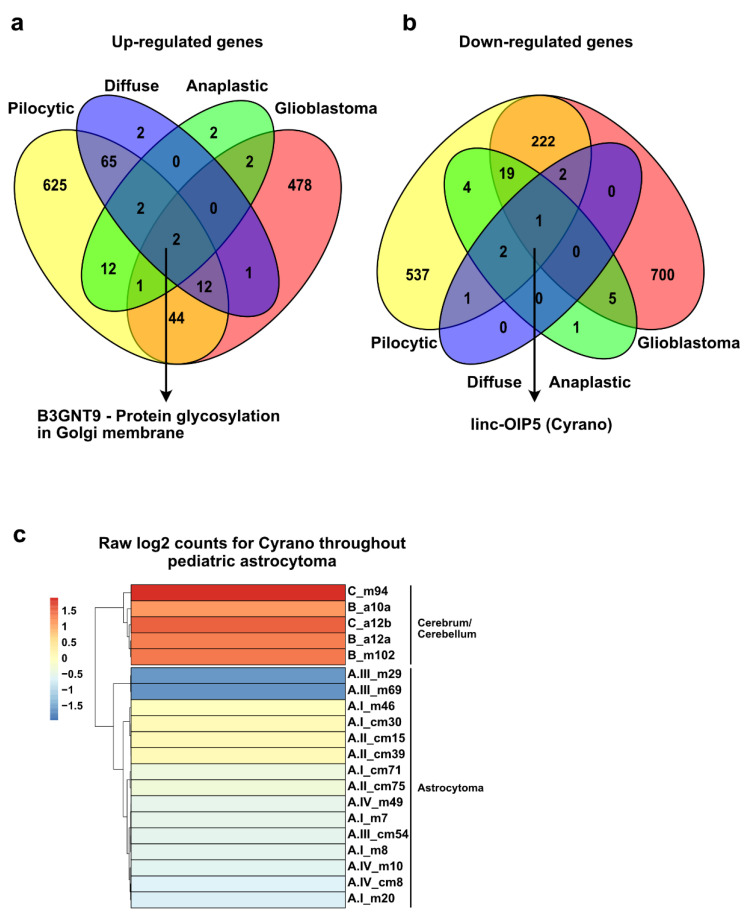
Shared misregulated genes among all the grades of pediatric astrocytoma. (**a**,**b**) Venn diagrams showing overlap in the number of up- (**a**) and downregulated genes (**b**) in the four grades of pediatric astrocytoma compared to healthy matching tissue. (**c**) Expression of the lncRNA Cyrano throughout pediatric astrocytoma and healthy cerebellum/cerebrum. Color bar displays the raw log2 counts for Cyrano in our RNA-seq data.

**Figure 7 ijms-23-12696-f007:**
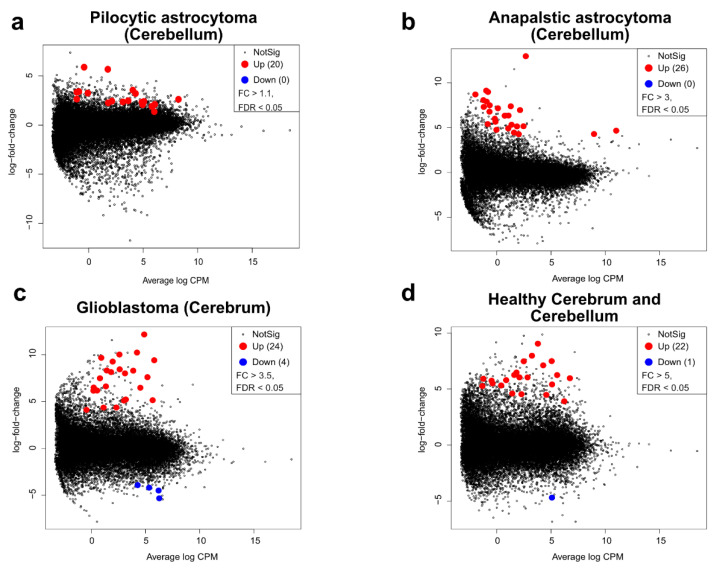
Uniquely expressed genes in pediatric astrocytoma. (**a**–**d**) Mean-difference plots highlighting the upregulated (red) and downregulated (blue) genes in pilocytic astrocytoma in the cerebellum (**a**) (FC > 1.1, FDR < 0.05), anaplastic astrocytoma in the cerebellum (**b**) (FC > 3, FDR < 0.05), glioblastoma in the cerebrum (**c**) (FC > 3.5, FDR < 0.05), and in healthy cerebrum and cerebellum (**d**) (FC > 5, FDR < 0.05). Comparisons were made with all the remaining grades of astrocytoma and healthy control tissue.

**Figure 8 ijms-23-12696-f008:**
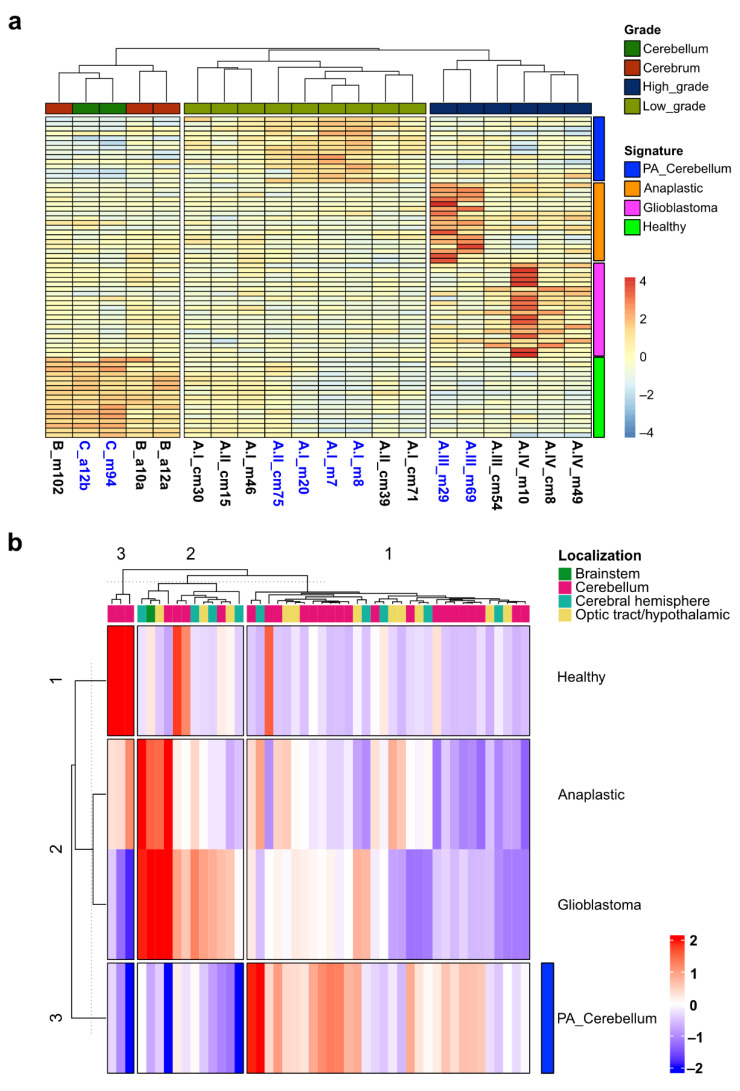
Clustering of pediatric astrocytomas according to their uniquely expressed markers. (**a**) Heat map displaying the hierarchical clustering of low- and high-grade pediatric astrocytoma and healthy cerebrum/cerebellum based on a panel of uniquely expressed genes. (**b**) Heatmap with hierarchical clustering of the ssGSEA enrichment scores for the genes in the transcriptional signature projected in the publicly available GEO dataset GSE73066 with the transcriptional profile of 47 pediatric PAs in different locations of the brain.

## Data Availability

All RNA-seq data were deposited on the GEO portal under the accession number GSE196694. Genes from the gene signature panel can be obtained upon request stating their use only for academic, nonprofitable studies. The custom code used in this work did not differ significantly from the implementation referenced.

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
