# Peer review of "The Transcriptomic Landscape of Pediatric Astrocytoma"

_ijms, 2022, doi:10.3390/ijms232012696_

Round 1

Reviewer 1 Report

Dear Authors,

Overall this is an interesting manuscript, with few minor shortcomings that need to be addressed. 

There is an incomplete sentence on line 10 as follows: "Percentages expected for libraries of good quality [10].", please reformat and/or expand.

LInes 105-107: it is not clear how the clustering was performed. Did you use reads or genes reads were mapped onto? If you used unprocessed reads, as you are stating, please provide more detailed explanation on the procedure.

LInes 117-118: You mention that you performed unsupervised clustering on the 500 most variable genes, while above you mentioned unprocessed reads? This leads me to the following issue, Figure 2a and 2b displaying PCA of the astrocytoma RNA-seq data needs to be described in the text with a sentence or two explaining the exact nature of the RNA-seq data used in the aforementioned analysis.

Lines 152-155: There is a typo (please use "studies" instead of "work") and it is an overstatement to say that this validates your data due to relatively limited sample size. 

Figure 3a and 3b: bad quality of pie charts.

Table 1 should be more nicely formatted in order to clearly separate each top 10 group.

LInes 233-241: Again, an overstatement to claim data correlation and validation since low quality RNA isolated from paraffin embedded (FFPE) tissue is often partially degraded and has a low overall quality.

Figure 5 - bad quality of pie charts

LInes 333-337: there is a typo - omit "with" on line 335

Line 390 typo: replace "envison" with "envisioned"

LIne 399 typo: replace "replicated" with "replicating"

Author Response

Reviewer 1

Dear Authors,

Overall this is an interesting manuscript, with few minor shortcomings that need to be addressed. 

There is an incomplete sentence on line 10 as follows: "Percentages expected for libraries of good quality [10].", please reformat and/or expand.

R: We have expanded the sentence.

Lines 105-107: it is not clear how the clustering was performed. Did you use reads or genes reads were mapped onto? If you used unprocessed reads, as you are stating, please provide more detailed explanation on the procedure.

R: We have clarified this by stating the use of mapped reads

Lines 117-118: You mention that you performed unsupervised clustering on the 500 most variable genes, while above you mentioned unprocessed reads? This leads me to the following issue, Figure 2a and 2b displaying PCA of the astrocytoma RNA-seq data needs to be described in the text with a sentence or two explaining the exact nature of the RNA-seq data used in the aforementioned analysis

R: By clarifying the previous comment we also solved this issue. We have homogenized the information in main text and figure legend to state correctly that we used RNA-seq data (mapped reads)

Lines 152-155: There is a typo (please use "studies" instead of "work") and it is an overstatement to say that this validates your data due to relatively limited sample size.

R: We have fixed this typo and step down the previous overstatement.

Figure 3a and 3b: bad quality of pie charts

R: The issue was solved by submitting the high-resolution images separately from this revised version of the ms.

Table 1 should be more nicely formatted in order to clearly separate each top 10 group.

R: The table has been adjusted as suggested by the reviewer and also sent to the supplementary data as suggested by the other reviewer.

Lines 233-241: Again, an overstatement to claim data correlation and validation since low quality RNA isolated from paraffin embedded (FFPE) tissue is often partially degraded and has a low overall quality.

R: We have tune down this overstatement

Figure 5 - bad quality of pie charts

R: The issue was solved by submitting the high-resolution images separately from this revised version of the ms.

Lines 333-337: there is a typo - omit "with" on line 335

R: We have fixed this typo.

Line 390 typo: replace "envison" with "envisioned"

R: We have fixed this typo.

LIne 399 typo: replace "replicated" with “replicating"

R: We have fixed this typo.

Reviewer 2 Report

Hernandez et al. presented a transcriptomic landscape for pediatric astrocytoma in this manuscript. Using the data from 15 biopsies and 5 matched normal tissues, they clustered the gene expression profiles based on tumor grades. To characterize the pattern, they identified the differentially expressed genes (DEGs) for each grade by comparing them with associated normal tissue. The DEGs analysis is further used to highlight pathway enrichment for each astrocytoma grade. Further, they compare the DEGs across different grades. Based on the unique up-regulated genes across each grade, they provide a transcriptional signature for astrocytoma grade-based classification. This study adds to the pediatric astrocytoma classification. Interestingly they recapitulated the heterogenous pattern among low-grade astrocytoma concerning their spatial locations mentioned by previous studies. The transcriptional signature for pediatric astrocytoma classification mapping with the tumor grade classification is also interesting. Several technical and conceptual issues should be addressed before publication. 1. The DEGs analysis needs further clarification. Since astrocytoma has been divided transcriptionally into low and high grades. Also, the authors highlight that the spatial location of the tumor adds a difference to DEGs analysis. Then it is also important to identify the DEGs by comparing high versus low-grade transcriptional profiles. Further, the DEGs need to be compared with previous astrocytoma classifications as mentioned by various studies (Zakrzewski et al., 2015, BMC Cancer; Rorive et al., 2006, Journal of Neuropathology & Experimental Neurology; Sharma et al., 2007, Cancer Research; Alige et al., 2014, Plos One). All these previous classifications should also be added to the references and discussion section. 2. Based on the DEGs analysis, the authors defined the transcriptional signature for tumor grade classification. The signature is differentiating the cerebrum and cerebellum highgrade tumors quite well, but the low-grade tumors and normal tissues show no distinct pattern. Although there are minor differences, the pattern is not as strong as in highgrade classes. It is important to reconsider the signature definition based on the revised DEGs analysis as suggested in step 1. Further, it is important to test this signature in an independent data set to highlight its robustness. It would be useful to use some signature scoring methods like ssGSEA or GSEA to test the signature and highlight the statistical differences among tumor grades. 3. Discussion section of signature classification provides references for breast cancerrelated signatures. It is important to highlight the previous astrocytoma classifications or related brain tumor classifications in this section. 4. Introduction should add more references to the previous related studies focusing on transcriptional classifications. 5. Some minor details, All the figures should highlight the tumors with their sample IDs as in Figures 2c and 7e it is difficult to separate which normal or which tumor case authors are referring to. 6. It is better to represent the tables in the main text in the form of bar plots or move them to the supplement section.

Author Response

Hernandez et al. presented a transcriptomic landscape for pediatric astrocytoma in this manuscript. Using the data from 15 biopsies and 5 matched normal tissues, they clustered the gene expression profiles based on tumor grades. To characterize the pattern, they identified the differentially expressed genes (DEGs) for each grade by comparing them with associated normal tissue. The DEGs analysis is further used to highlight pathway enrichment for each astrocytoma grade. Further, they compare the DEGs across different grades. Based on the unique up-regulated genes across each grade, they provide a transcriptional signature for astrocytoma grade-based classification. This study adds to the pediatric astrocytoma classification. Interestingly they recapitulated the heterogenous pattern among low-grade astrocytoma concerning their spatial locations mentioned by previous studies. The transcriptional signature for pediatric astrocytoma classification mapping with the tumor grade classification is also interesting. Several technical and conceptual issues should be addressed before publication.

  1. The DEGs analysis needs further clarification. Since astrocytoma has been divided transcriptionally into low and high grades. Also, the authors highlight that the spatial location of the tumor adds a difference to DEGs analysis. Then it is also important to identify the DEGs by comparing high versus low-grade transcriptional profiles. Further, the DEGs need to be compared with previous astrocytoma classifications as mentioned by various studies (Zakrzewski et al., 2015, BMC Cancer; Rorive et al., 2006, Journal of Neuropathology & Experimental Neurology; Sharma et al., 2007, Cancer Research; Alige et al., 2014, Plos One). All these previous classifications should also be added to the references and discussion section.

R: We appreciate and recognize the insights that this and the next comment added to our work. By reinforcing the transcriptional signature using the suggested method (ssGSEA), we think that condensing four grades into low- and high-grade is not essential. Thus, although it would be very simple to do, we do not incorporate information about DEGs between low and high degree. See the answer to the next comment for further clarification.

Regarding the comparison of DEGs with previous astrocytoma classifications, we did only those that performed the same analysis than us (e.g., pediatric astrocytoma versus matching healthy tissue as in Rorive et al., 2006, Journal of Neuropathology & Experimental Neurology) and added this information in the manuscript, references and supplementary figure 4. However, we used the suggested studies (Zakrzewski et al., 2015, BMC Cancer, Sharma et al., 2007, Cancer Research) to test or gene signature. All the suggested studies have now been incorporated in the results and discussion sections.

  1. Based on the DEGs analysis, the authors defined the transcriptional signature for tumor grade classification. The signature is differentiating the cerebrum and cerebellum highgrade tumors quite well, but the low-grade tumors and normal tissues show no distinct pattern. Although there are minor differences, the pattern is not as strong as in highgrade classes. It is important to reconsider the signature definition based on the revised DEGs analysis as suggested in step 1. Further, it is important to test this signature in an independent data set to highlight its robustness. It would be useful to use some signature scoring methods like ssGSEA or GSEA to test the signature and highlight the statistical differences among tumor grades.

R: We have performed the suggested scoring method ssGSEA to test the signature. Furthermore, with the same method of ssGSEA we have tested our signature in two independent data sets of pediatric pilocytic astrocytoma. This information has been added in the main results with a new figure. Additionally, we have added some results regarding the cerebellar PA that may further support our gene signature specific for this type of astrocytoma.

  1. Discussion section of signature classification provides references for breast cancerrelated signatures. It is important to highlight the previous astrocytoma classifications or related brain tumor classifications in this section.

R: Previous astrocytoma classifications and gene signatures have been incorporated.

  1. Introduction should add more references to the previous related studies focusing on transcriptional classifications.

R: Previous astrocytoma classifications and gene signatures have been incorporated mainly in the discussion section.

  1. Some minor details, All the figures should highlight the tumors with their sample IDs as in Figures 2c and 7e it is difficult to separate which normal or which tumor case authors are referring to.

R: Fixed

  1. It is better to represent the tables in the main text in the form of bar plots or move them to the supplement section.

R: Tables have been moved to the supplement section

Round 2

Author Response

We would like to thank once again the reviewer for his/her constructive comments, which have greatly contributed to improve the quality of the manuscript. Below you will also find a response to the reviewer comments.

  1. Testing of the transcriptional signature in their own data set is not required since the signature is derived from the same data. It is therefore suggested to move Figure 8a to the supplement.

  1. Figure has been moved to Supplementary Figure 5a

  1. Based on the signature testing in the GSE73066 data set, the performance of the signature is not robust. Since all the cases are pilocytic astrocytoma, then the percentage of tumors (Cluster 1,2, and 4) enriched for glioblastoma, and anaplastic astrocytomas are way higher than PA astrocytomas. It indicates that the signature of PA astrocytoma needs further improvement. Similarly, signature testing in GSE5675 clearly indicates that the PA signature shows high enrichment with the GBM and anaplastic signatures for most of the cases. And this data set is also PA based dataset.

  1. Signature testing in both the independent data sets clearly reflects that PA signature needs further improvement.

  1. Authors mentioned that the PA cerebellum and cerebrum are different in their cohort due to the presence of fusion. Figure 3e shows an equal number of cases with and without fusion. Also, the number of cerebrum comparison with normal doesn’t generate any DEGs. It would be important to highlight the DEGs in PAs among these two regions of the brain following their genetic differences. This would help to improve the PA signature. Further pairwise comparisons of PA with anaplastic and GBM tumors would highlight more specific and distinct gene features. It is therefore suggested to improve the signature and then retest it in the independent datasets

R: Comments 2-4 have been addressed as a single one. Comparison between DEGs in PA from the two different regions of the brain has been done. Although no DEGs were found, this information has been incorporated.

Additionally, to further refine the cerebellar PA signature we increased the genes of the panel by adding 1) 10 unique genes with a slighter lower fold change, and 2) 7 genes that were found by direct pairwise comparison of the PA versus each of the other types of astrocytoma. Direct pairwise comparisons only detected DEGs between cerebellar PA vs Glioblastoma. We have used this new refined gene signature panel to test it on the independent databases as suggested by the reviewer.

Round 3

Reviewer 2 Report

Hernandez et al. have addressed all the major concerns related to their transcriptomic landscape of pediatric astrocytoma.